# Identification and Characterization of a Novel *N*- and *O*-Glycosyltransferase from *Saccharopolyspora erythraea*

**DOI:** 10.3390/molecules25153400

**Published:** 2020-07-27

**Authors:** Fabienne Gutacker, Yvonne-Isolde Schmidt-Bohli, Tina Strobel, Danye Qiu, Henning Jessen, Thomas Paululat, Andreas Bechthold

**Affiliations:** 1Institute for Pharmaceutical Biology and Biotechnology, Albert-Ludwigs-University, Stefan-Meier-Straße 19, 79104 Freiburg, Germany; fabienne.gutacker@pharmazie.uni-freiburg.de (F.G.); yvonne.schmidt@live.de (Y.-I.S.-B.); tinastrob@aol.com (T.S.); 2Institute of Organic Chemistry, Albert-Ludwigs-Universität, Albertstrasse 21, 79104 Freiburg, Germany; danye.qiu@oc.uni-freiburg.de (D.Q.); henning.jessen@oc.uni-freiburg.de (H.J.); 3Organic Chemsitry II, Universität Siegen, Adolf-Reichwein-Strasse 2, 57068 Siegen, Germany; paululat@chemie.uni-siegen.de

**Keywords:** glycosyltransferase, glycohydrolase, *Saccharopolyspora erythraea*, anthraquinone, glycobiology, nucleotide-activated sugar donor

## Abstract

Glycosyltransferases are important enzymes which are often used as tools to generate novel natural products. In this study, we describe the identification and characterization of an inverting *N*- and *O*-glycosyltransferase from *Saccharopolyspora erythraea* NRRL2338. When feeding experiments with 1,4-diaminoanthraquinone in *Saccharopolyspora erythraea* were performed, the formation of new compounds (U3G and U3DG) was observed by HPLC-MS. Structure elucidation by NMR revealed that U3G consists of two compounds, *N*_1_-α-glucosyl-1,4-diaminoanthraquinone and *N*_1_-β-glucosyl-1,4-diaminoanthraquinone. Based on UV and MS data, U3DG is a *N*_1_,*N*_4_-diglucosyl-1,4-diaminoanthraquinone. In order to find the responsible glycosyltransferase, gene deletion experiments were performed and we identified the glycosyltransferase Sace_3599, which belongs to the CAZy family 1. When *Streptomyces albus* J1074, containing the dTDP-d-glucose synthase gene *oleS* and the plasmid pUWL-A-*sace_3599*, was used as host, U3 was converted to the same compounds. Protein production in *Escherichia coli* and purification of Sace_3599 was carried out. The enzyme showed glycosyl hydrolase activity and was able to produce mono- and di-*N*-glycosylated products in vitro. When UDP-α-d-glucose was used as a sugar donor, U3 was stereoselective converted to *N*_1_-β-glucosyl-1,4-diaminoanthraquinone and *N*_1_,*N*_4_-diglucosyl-1,4-diaminoanthraquinone. The use of 1,4-dihydroxyanthraquinone as a substrate in in vitro experiments also led to the formation of mono-glucosylated and di-glucosylated products, but in lower amounts. Overall, we identified and characterized a novel glycosyltransferase which shows glycohydrolase activity and the ability to glycosylate “drug like” structures forming *N*- and *O*-glycosidic bonds.

## 1. Introduction

Natural products continue to serve as a key platform for drug development and it is well established that glycosylation of small molecules can dramatically influence the pharmacological properties of the parent scaffold. In glycosylated natural products, the sugar moieties are often crucial for activity and it has been shown that sugar moieties of a natural compound directly interact with the cellular targets. Today glycosylated natural products are often used as antibiotics (e.g., aminoglycosides, macrolides like erythromycin), drugs with antifungal activities (e.g., polyenes like amphotericin and nystatin) and cytotoxic drugs (e.g., angucyclines like doxorubicin) [1,2,3,4,5,6,7,8].

Catalytic modules, which degrade, modify, or create glycosidic bonds are called Carbohydrate Active enZymes and are grouped in the CAZy database (http://www.cazy.org/). CAZy families are classified according to amino acid sequence similarities [9]. Enzymes belonging to the superfamily of glycosyltransferases (GTs) catalyze the transfer of sugar moieties from activated sugar donors onto acceptor molecules by forming glycosidic bonds [1,10,11]. GTs are distributed in both prokaryotes and eukaryotes, and demonstrate enormous structural and functional variety [12,13]. Acceptor molecules can be structurally versatile, comprising sugars, lipids, nucleic acids, steroid hormones, biogenic amines, secondary metabolites from plants and bacteria, and therapeutic drugs [11].

*N*-glycosylation is a ubiquitous modification that occurs mainly in proteins. This process involves the transfer of preassembled oligosaccharides from a lipid donor to asparagine side chains of polypeptides and is catalyzed by membrane-bound oligosaccharyltransferase (OST) [14]. A few nitrogen glycosylating GTs (*N*-GTs) have been identified that use nucleotide-activated monosaccharides as donors to modify either asparagine residues of peptides and proteins or to introduce sugar moieties to natural products. For example, the *N*-GT PtmJ from *Streptomyces pactum* is involved in biosynthesis of the antitumor antibiotic pactamycin [15] and AaNGT from *Aggregatibacter aphrophilus* is able to utilize different sugar donors [16]. Further examples are Asm25, a *N*-GT involved in ansamitocin biosynthesis [17] and StnG that glycosylates the imine nitrogen atom of guanidine during streptothricin biosynthesis [18]. As indicated above, attachment or exchange of sugar moieties can essentially optimize the pharmacological characteristics of a drug and modify the molecular mode of action [19,20,21]. There is a vivid demand in efficient procedures to alter the glycosylation pattern of diverse chemicals and, therefore, improve therapeutic agents. GTs can be used to perform these modifications in an effective and economic way in contrast to challenging chemical synthesis [22,23]. GT genes have been cloned and characterised to develop in vitro and in vivo techniques to glycosylate given compounds and generate new glycosylated bioactive compounds [24,25,26,27,28,29,30]. Previously, the *O*-glycosylation of a range of complex natural products and drugs with the help of these potent tools has been reported [31,32,33]. While such approaches have dramatically extended the structural diversity of a range of therapeutically important natural products, they have largely been restricted to *O*-glycosylation and, only to a lesser extent to *N*-glycosylation [34].

Actinomycetes are known for their ability to glycosylate several natural products. In order to find new, unknown GTs our working group started to screen different strains performing in vivo feeding experiments. *Saccharopolyspora erythraea* NRRL2338 is a Gram-positive mycelial soil Actinomycete, producing the clinically useful broad-spectrum macrolide antibiotic erythromycin A [35]. In this paper, we report the identification of a *N*- and *O*-GT of the CAZy family 1 from *Saccharopolyspora erythraea* NRRL2338 which is able to convert 1,4-diaminoanthraquinone (U3) to its *N*_1_**-β-glycosylated, *N*_1_-α-glycosylated and di-glycosylated derivatives. Moreover, it catalyzes the conversion of 1,4-dihydroxy-anthraquinone (U2) to its mono- and di-glucosylated products (U2G, U2DG).

## 2. Results

### 2.1. In Vivo Biotransformation of Anthraquinone Derivatives

To examine the biotransformation ability of *S. erythraea*, in vivo feeding experiments were performed. Therefore, 1,2-dihydroxyanthraquinone, 1,4-dihydroxyanthraquinone (U2), 1,4-diamino-anthraquinone (U3), 1,5-diaminoanthraquinone, 1-amino-2-methylanthraquinone, 1-amino-2-carboxyanthraquinone, 1-amino-2-chloroanthraquinone, and 1-amino-4-chloranthraquinone were dispensed to *S. erythraea* under whole cell feeding conditions. The crude extract of the cultures fed with U3 exhibited new natural compounds in HPLC-MS analysis at retention times of 4.2 min (U3DG) and 11.7 min (U3G) (Figure 1a I). In comparison to the MS spectrum of the substrate U3 *m/z* = 237.1 [M − H]^−^ for the new detected compound U3G MS signals of *m/z* = 399.2 [M − H]^−^, *m/z* = 435.2 [M + Cl]^−^ and *m/z* = 459.2 [M + CH_3_COO]^−^ were measured. For U3DG, MS signals of *m/z* = 561.2 [M − H]^−^ and *m/z* = 597.2 [M + Cl]^−^ were detected (Figure 1c). Feeding of the other anthraquinone derivatives did not result in the formation of any new compounds.

### 2.2. Genes Encoding for Glycosyltransferases in the Genome of S. erythraea

The whole genome of *S. erythraea* NRRL2338 was sequenced in 2007 [36], revealing 7198 protein-encoding sequences. Altogether 56 GT genes are present. GTs supposed to be involved in primary metabolism are responsible for cell wall synthesis and DNA formation. *S. erythraea* possesses also GTs participating in secondary metabolism. These enzymes generally have protein sequence homology to GTs of CAZy family 1. Three GT genes can be found in the genome of *S. erythraea* whose gene products show homology to Uridine diphosphate (UDP)-glucosyltransferases and *N*-GTs of family 1 (*sace_1884*, *sace_3599*, *sace_4470*). All these GT genes are not located close to biosynthetic gene clusters, but distributed throughout the genome.

### 2.3. Heterologous Expression of Sace_3599 in S. albus Gluc

To identify the *N*-GT responsible for *N*-glycosylation of U3 *S. albus* Gluc was generated. This strain contains *oleS*, a gene encoding a dTDP-d-glucose synthase, which is able to convert α-glucose 1-phosphate to dTDP-d-glucose as a putative sugar donor for the *N*-GT. Expression of *oleS* in the marker-free host strain *S. albus* Gluc is controlled by the constitutive *ermE** promoter [37]. Plasmids containing *sace_1884*, *sace_3599*, and *sace_4470* were introduced into *S. albus* Gluc, leading to *S. albus* Gluc x pUWL-A-*sace_1884*, *S. albus* Gluc x pUWL-A-*sace_3599*, and *S. albus* Gluc x pUWL-A-*sace_4470*, respectively. The compound U3 was fed to *S. albus* WT, *S. albus* Gluc, and all mutant strains containing one of the GT-genes from *S. erythraea*. Extracts were analysed by HPLC-MS. The crude extract of *S. albus* Gluc x pUWL-A-*sace_3599* exhibited a new peak at a retention time of 11.7 min (U3G) shown in Figure 1a VIII, while the other generated strains did not produce any new compounds. After solid phase fractionation of the extract using an Oasis HLB column, preparative thin-layer chromatography and Sephadex LH20 column, U3DG was also detected at 4.2 min by HPLC-MS (Figure 1a IV).

### 2.4. Gene Deletion and Complementation Experiments

In order to prove that *sace_3599* is the only GT of *S. erythraea*, responsible for the formation of the *N*-glucosylated derivatives of U3, gene inactivation experiments were carried out. A double-crossover mutant was generated in which *sace_3599* was replaced by a spectinomycin resistance-conferring gene. *S. erythraea* Δ*sace_3599* was not able to glucosylate U3 (Figure 1a V). Complementation of *S. erythraea* Δ*sace_3599* with an unscathed copy of the gene *sace_3599* restored biotransformation ability, indicating that Sace_3599 is the only *N*-glucosylating GT (Figure 1a VI).

### 2.5. Structure Elucidation of U3G and U3DG

U3G and U3DG were isolated from 1.5–5.l of *S. erythraea* and *S. albus* Gluc x pUWL-A-*sace_3599*, respectively, both fed with U3 and from in vitro glycosylation activity assays. Structure elucidation was performed using HPLC-MS and NMR spectroscopy. U3G consists of the *N*_1_-glucosylated derivative of the substrate U3. When UDP-α-d-glucose was used as a sugar donor, U3 was stereoselective converted in vitro to *N*_1_-β-glucosyl-1,4-diaminoanthraquinone. U3G isolated from Actinomycetes after whole-cell feeding experiments consisted of the *N*_1_-α-glucosylated derivative as well as the *N*_1_-β-glucosylated derivative (Figure 2). NMR-data are shown in Appendix A.

The percentage ratios of α- and β-anomer in U3G varied. Both anomers could be separated by capillary electrophoresis (Figure 3). Due to low amounts and purities, the structure of U3DG could not be clearly elucidated, but UV spectra, mass spectrometry and NMR data indicate, that two sugars are attached to the diaminoanthraquinone moiety via *N*-glycosylation. There is no evidence for an intact disaccharide moiety. Therefore, we conclude that the compound is *N*_1_,*N*_4_-diglucosyl-1,4-diaminoanthraquinone.

### 2.6. Production, Purification and In Vitro Activity Assay of Sace_3599

For further characterization of the identified GT Sace_3599, we decided to perform in vitro activity assays. Therefore, the gene *sace_3599* was expressed in *E. coli* Rosetta™. Soluble protein was obtained and purified as described in Materials and Methods (see Section 4.11). The molecular mass of the enzyme Sace_3599 was 45.3 kDa. A SDS Gel after immobilized-metal affinity chromatography (IMAC) with Ni^2+^-NTA and a gel filtration chromatogram can be found in Appendix A.

In in vitro activity assays, the highest conversion catalyzed by Sace_3599 was reached using TRIS buffer pH 8.8 and an incubation of 22 h at 28 °C. Although the enzyme was active without the addition of any metal ion, high activity was detected when Mn^2+^ or Mg^2+^ were added (Appendix A). In the presence of UDP-α-d-glucose, Sace_3599 converted U3 in in vitro assays to its *N*_1_-β-glucosylated derivative (U3G) which was detected by HPLC-MS and shown by NMR spectroscopy. In addition, we observed the formation of the *N*_1_,*N*_4_-diglucosylated derivative (U3DG, Figure 4a III; Figure 4b). In negative controls without Sace_3599, only U3 and no biotransformation products were detected. The enzyme also accepted TDP-α-d-glucose as a sugar donor in small quantities (Figure 4a I, Figure 4b), but not UDP-α-d-galactose (Figure 4a II; Figure 4b).

Next, in order to test, whether Sace_3599 can also form *O*-glycosidic bonds, 1,4-dihydroxyanthraquinone (U2) was tested as a substrate. In the presence of UDP-α-d-glucose, Sace_3599 converted 1,4-dihydroxyanthraquinone (U2, at 29.3 min) to two new products at retention times of 4.2 min (U2DG) and 16.0 min (U2G, Figure 5b I). Due to low amounts and purities, the structure of U2G and U2DG could not be clearly elucidated, but UV spectra (Figure 5c) and mass spectrometry (Figure 5d) indicate that compound U2G is *O*_1_-glucosyl-1,4-dihydroxyanthraquinone and U2DG is *O*_1_,*O*_4_-diglucosyl-1,4-dihydroxyanthraquinone (Figure 5a). For U2G, characteristic MS signals of *m/z* = 401.1 [M − H]^−^ and *m/z* = 437.1 [M + Cl]^−^ were detected. U2DG is less pure but characteristic MS signals of *m/z* = 563.2 [M − H]^−^, *m/z* = 599.2 [M + Cl]^−^ and *m/z* = 623.3 [M + CH_3_COO]^−^ were detected. In negative controls without Sace_3599, no biotransformation products of U2 were detected by HPLC-MS analysis.

In order to test whether Sace_3599 exhibits glycohydrolase activity UDP-[^14^C]glucose was added as substrate to in vitro assays with 50 µM Sace_3599. After 1, 2 and 22 h samples of 1 µL were taken for thin layer chromatography (see Section 4.13). Interestingly, a lot more free [^14^C]glucose in the assay-samples was detected than in the negative controls without enzyme (Figure 6).

### 2.7. Genome Sequence Analysis

In silico analysis of a 10.5 kb DNA fragment containing *sace_3599* revealed that the gene is enclosed by genes encoding for proteins with unknown function, named hypothetical proteins (*hp*), a transcriptional Xenobiotic Response Element regulator (*xre-reg*), a polymorphic GC-repetitive sequence (*pe-pgrs*) family protein, a methionine aminopeptidase (*meth-ap*), two aminoglycoside N3′ acetyltransferases (*ag-at*), an oxidase (*ox*), a class II aldolase (*cII-ald*), and an enolase-phosphatase (*en-phos*), respectively (Figure 7).

### 2.8. Homologous Proteins to Sace_3599

Protein BLAST [38] showed high sequence identity of Sace_3599 to other known glycosyltransferases which are involved into antibiotic metabolism or antibiotic biosynthetic process like YjiC (*Bacillus subtilis*) [39,40], OleI (*Streptomyces antibioticus*) [41,42,43], TylCV (*Streptomyces fradiae*) [44], OleD (*Streptomyces antibioticus*) [41,42,43], Mgt (*Streptomyces lividans*) [45], YojK (*Bacillus subtilis*) [40,46] and YdhE (*Bacillus subtilis*) [40,46,47]. YjiC, an uncharacterized UDP-glucosyltransferase and OleI are the most similar GTs with 39% identical amino acids. OleI, TylCV (38% identity), OleD (34% identity) and Mgt (34% identity) are Macrolide glycosyltransferases. Sequence alignment is shown in Figure 8. Similar amino acids are highlighted in grey, amino acids binding the sugar donor in OleI are marked with blue arrows, while amino acids binding the acceptor substrate in OleI are marked with turquoise arrows [42]. All homologous GTs belong to CAZy family 1. This family mainly consists of inverting GTs belonging to GT-B superfamily [9]. In contrast to GTs belonging to GT-A superfamily, GT-B-GTs do not contain a conserved DxD motif for coordination of divalent metal ions (usually Mg^2+^ or Mn^2+^) [11].

## 3. Discussion

Anthraquinones and their derivatives have diverse physiological and pharmacological properties including antibiotic, antifungal [48] and anticancer activities [49,50]. Saccharide moieties are, in many cases, essential for the biological activity of glycosylated secondary metabolites. Conjunctions of sugars usually result in improved activities or properties as, for example, better solubility, increased polarity, and improved chemical stability [51]. GTs, which catalyze the conjunction of a sugar moiety with an aglycon, are main enzymes for generating novel compounds with changed glycosylation patterns.

We aimed to find GTs that catalyze exceptional sugar attachment reactions. Therefore, we became interested in *Saccharopolyspora erythraea*. After feeding experiments with various types of anthraquinones, in silico analysis and knockout experiments, we identified Sace_3599 to be an *N*-GT from CAZy family 1. Further biotransformation experiments in *S. erythraea*, heterologous expression in *S. albus* Gluc and in vitro glycosylation assays pointed out that Sace_3599 catalyzes the formation of mono- and di-glucosylated derivatives of 1,4-Diaminoanthraquinone (U3). When U3 together with UDP-α-d-glucose as a sugar donor was used in in vitro experiments, the formation of *N*_1_-β-glucosyl-1,4-diaminoanthraquinone was detected by UV spectra, mass spectrometry, and NMR data. This result indicates that Sace_3599 may be a GT with an inverting reaction mechanism as published by Lairson et al. [11], which was already proposed by CAZy database [9].

In vitro assays have also shown that Sace_3599 is more active at higher pH values and that the GT shows high selectivity towards the sugar donor. This was shown for GTs like OleD (*Streptomyces antibioticus*), Mgt (*Streptomyces lividans*) and others before [42,45,52]. When no divalent cations as cofactors were added, Sace_3599 was still active, which supports the assignment to the GT-B superfamily through the CAZy database [9]. When 1,4-Dihydroxyanthraquinone (U2) was used as substrate, two new, more hydrophilic peaks at retention times of 16.0 min (U2G) and 4.2 min (U2DG) appeared in HPLC/MS analysis which were not present in negative controls without Sace_3599. UV spectra and mass spectrometry indicated, that U2G is the mono-glucoside and U2DG is the di-glucoside of U2. This result leads to the assumption, that Sace_3599 is quite flexible towards the acceptor substrate and that Sace_3599 also functions as an *O*-GT. Other GTs have been published, which are also able to form different glycosidic bonds. This property makes them quite attractive for both research and industry. One example is the GT UGT71E5 from *Carthamus tinctorius*, which catalyzes the formation of *N*-, *O*- and *S*-glycosidic bonds [53]. To test the *S*-glycosylation activity of Sace_3599, in vitro assays with acceptor substrates like 7-Mercapto-4-methylcoumarin or 3,4-dichlorothiophenol could be carried out.

In order to test the glycohydrolase activity of Sace_3599, in vitro activity assays with UDP-[^14^C]glucose were carried out following a protocol by Levanova et al. [54]. Since a lot more of free [^14^C]glucose was detected in the assay-samples compared to the negative controls, Sace_3599 seems to have some glycohydrolase activity in addition to the glycosyltransferase activity. Although this result is not surprising it is worth mentioning that there are other GTs like lymphocyte surface or ecto-GTs which do not hydrolase the nucleotide sugars [55].

After feeding the “drug like” structure 1,4-U3 to bacterial strains expressing *sace_3599*, we interestingly observed the formation of the *N*_1_-α-glucosylated derivative and the *N*_1_-β-glucosylated derivative. The percentage ratios of the anomers varied, probably depending on the growth stages of the strains. Since there was no glycosylated U3 in biotransformation experiments with *S. erythraea* Δ*sace_3599* or *S. albus* Gluc detectable, Sace_3599 is the only GT in our Actinomycetes which glycosylates U3. Mutarotation, known from glucose, might be a reason for the difference in the configuration of the anomeric centre. As mutarotation cannot occur in nucleotide sugars such as UDP-d-glucose, the only possibility where mutarotation could take place would be in the products. Mutarotation of secondary amines has been reported before [56]. During our study, we could not observe epimerization of C-1 in the sugars in these compounds even after prolonged storage. Moreover, both ROESY- and EXSY-NMR show no exchange signals between the two components. Even at different temperatures (25–80 °C), the ratio between the two components did not change (Appendix A). These results lead us to the assumption that the α- and β-glucosylated derivatives are stable without the occurrence of mutarotation. Another explanation for the formation of both anomers might be the existence of an epimerase in both bacterial strains, that catalyzes the epimerization of the mono-glucosylated product (U3G) or, we believe, Sace_3599 might show substrate flexibility towards the sugar donor substrate and accept UDP-α-d-glucose as well as UDP-β-d-glucose.

## 4. Materials and Methods

### 4.1. Bacterial Strains and Growth Conditions

*Saccharopolyspora erythraea* NRRL 2338 [57], *Streptomyces albus* J1074 [58], and their mutants were cultivated on mannitol soya flour agar plates [59] containing 10 mM MgCl_2_ or tryptic soya agar plates [60] at 28 °C for at least 48 h. The cultivation was carried out in tryptic soya broth (TSB) in double-baffled Erlenmeyer flasks with stainless steel spring at 28 °C and 180 rpm on a rotary shaking incubator for at least 48 h. For selection the medium was supplied with apramycin and spectinomycin to final concentrations of 100 µg/mL each. *Escherichia coli* Turbo was used for cloning, the strain ET12567 x pUZ8002 was used for the intergeneric conjugal transfer of non-methylated plasmid DNA from *E. coli* to the actinomycete recipient, and Rosetta™ 2 cells (41402 Novagen, Darmstadt, GER) were used as host for the heterologous expression of the Sace_3599 protein applying standard protocols [36,37,61]. The *E. coli* strains were grown on lysogeny broth (LB-Lennox) [62,63] agar plates at 37 °C or in LB-Lennox liquid medium at 20 °C to 37 °C, shaking at 180 rpm in baffled Erlenmeyer flasks. The media contained the appropriate selection antibiotic at a final concentration of 30 µg/mL kanamycin and 34 µg/mL chloramphenicol accordingly.

### 4.2. Plasmid Construction

Plasmids used in this study are listed in Table 1. Primer sequences and cloning protocols for the generation of the plasmids are delineated in Appendix A. All plasmids were confirmed by restriction endonuclease digestion and sequencing.

### 4.3. Whole-Cell Feeding Experiments

In order to determine the biotransformation abilities of the *S. erythraea* NRRL2338 WT as well as the mutants *S. albus* Gluc x pUWL-A-*sace_1884*, *S. albus* Gluc x pUWL-A-*sace_3599*, and *S. albus* Gluc x pUWL-A-*sace_4470*, feeding experiments were performed. A total of 100 mL of NL111 medium [71], containing the appropriate antibiotics, were inoculated with 1 mL of a 24-h-preculture of the respective actinomycete strain grown in TSB in a rotary shaking incubator at 28 °C and 180 rpm. After 36 h the culture was fed with 20 µL of test compound dissolved in dimethyl sulfoxide (DMSO) (0.1 M). The feeding procedure was performed five times in total at 12 h intervals. The culture was harvested (10 min, 1000 g) 7 days after inoculation and extracted with ethyl acetate in a ratio of 1:1 after adjustment to pH 7. The solvent was removed in vacuo. The received crude extract was analysed for biotransformation products by high-pressure liquid chromatography–mass spectrometry (HPLC-MS). 1,2-dihydroxyanthraquinone, 1,4-dihydroxyanthraquinone (U2), 1,4-diamino-anthraquinone (U3), 1,5-diaminoanthraquinone, 1-amino-2-methylanthraquinone, 1-amino-2-carboxyanthraquinone, 1-amino-2-chloroanthraquinone, and 1-amino-4-chloranthraquinone were used as test compounds.

### 4.4. Heterologous Test System

The plasmid pTOS-Gluc, containing the dTDP-d-glucose synthase gene *oleS*, was integrated into the genome of *S. albus*. Recombinant clones were isolated and used as host for transformation with pUWL-Dre capable of expressing the recombinase gene *dre* [61]. Expression of *dre* led to the marker-free mutant *S. albus* Gluc. This strain was used as host for transformation with *sace_1884*, *sace_3599* or *sace_4470*, which have been cloned into pUWL-A before.

### 4.5. Inactivation of Sace_3599 in the Genome of S. erythraea

The GT gene *sace_3599* was inactivated by the insertion of a resistance-conferring gene via homologous recombination. For that reason, two homologous regions of approximately 1.7 kb, including either a part of the beginning or the end of the gene, were amplified and cloned into the suicide vector pKC1132. In the middle of the gene a spectinomycin resistance-conferring gene was cloned. The plasmid pKCΔ*sace_3599* was conjugally transferred from *E. coli* ET12567 x pUZ8002 into *S. erythraea*. Single-crossover mutants were obtained overlaying the conjugation with apramycin and spectinomycin. After 10–12 passages, in liquid medium with spectinomycin single colonies were picked both on apramycin- or spectinomycin-containing agar plates. The required double-crossover mutants were screened for the loss of apramycin resistance. PCR was used to prove the successful inactivation of *sace_3599*. In order to determine the biotransformation abilities of the resulting mutant feeding experiments were performed.

### 4.6. Complementation of S. erythraea Δsace_3599 with an Intact Copy of Sace_3599

The plasmid pTOS(z)-*sace_3599* was conjugally transferred from *E. coli* ET12567 x pUZ8002 into *S. erythraea* Δ*sace_3599*. Feeding experiments were performed using the resulting spectinomycin- and apramycin-resistant mutant.

### 4.7. Purification of Biotransformation Products

Novel compounds were isolated from the crude extracts for structure elucidation. After fractionation of the extracts using Oasis HLB columns (Waters, Eschborn, Germany) applying a methanol/water gradient, preparative thin-layer chromatography (silica gel 60 F_254_, Macherey-Nagel, Düren, Germany) was developed using acetonitrile-water-acetic acid (89.4:10.5:0.1) or dichloromethane-methanol-acetic acid (89.9:10.0:0.1) as solvent. In the last step, the biotransformation products were purified by gelfiltration on Sephadex LH20 column (GE Healthcare, Solingen, Germany) with methanol. After each purification step, the solvent was removed in vacuo.

### 4.8. Sample Analysis by HPLC-MS

For analysis the dried samples were dissolved in methanol. HPLC-MS analysis was performed using an Agilent 1100 series LC/MS system (Agilent Technologies, Santa Clara, USA) with electrospray ionization (ESI) and detection in positive and negative modes. The LC system contained a Zorbax Eclipse XDB-C8 column (5 µm particle size; 150 mm × 4.6 mm; Agilent Technologies, Santa Clara, CA, USA) and a Zorbax XDB-C8 precolumn (5 µm particle size; 12.5 mm × 4.6 mm; Agilent Technologies, Santa Clara, CA, USA), kept at 18 °C. Detection wavelengths/reference wavelengths of the diode array were 254/360, 480/800, 300/450, 430/600 and 550/700 nm. Solvent A (acetonitrile) and solvent B (acetic acid (0.5 % *v/v*) in water) were used in the following gradient: 80 to 67% solvent B (0 to 9 min), 67 to 50% solvent B (9 to 16 min), 50 to 30% solvent B (16 to 20 min), 30 to 5% solvent B (20 to 24 min), a hold at 5% solvent B (24 to 30 min) and a hold at 80% solvent B (30 to 35 min) at a flow rate of 0.5 mL/min.

### 4.9. Structure Elucidation Using NMR Spectroscopy

Nuclear Magnetic Resonance spectra (NMR) of U3G and U3DG isolated from in vitro experiments with Sace_3599 and in vivo feeding experiments with *S. erythraea* and *S. albus* Gluc x pUWL-A-*sace_3599* were measured using a Varian VNMR-S 600MHz spectrometer equipped with 3 mm triple resonance inverse and 3 mm dual broadband probes. Pulse sequences were taken from Varian pulse sequence library. Spectra were recorded in 150 µL DMSO-d_6_ at 35 °C. Variable temperature experiments were measured at 25, 35, 50, 65 and 80 °C in DMSO-d_6_. EXSY spectra were recorded using the NOESY pulse sequence with a mixing time of 3 s.

### 4.10. Separation of Mono-Glucosylated Products Using Capillary Electrophoresis

Capillary electrophoresis (CE) with U3G of extracts from *S. erythraea* and *S. albus* Gluc x pUWL-A-*sace_3599* and the products of the in vitro reactions were performed. Therefore, a fused silica capillary with 40 cm effective length and 50 μm inner diameter was used. Running buffer consisted of 20 mM Borate, pH 9.3 with 50 mM SDS. Samples were injected at 100 mbar for 5 s. Separation voltage of 15 kV, temperature of 20 °C and UV detection at 254 nm were used.

### 4.11. Expression of Sace_3599 in E. coli and Purification of the Protein

For heterologous expression of *sace_3599* the plasmid pET28a-*sace_3599*-N-his_6_ was constructed. Rosetta™ 2 cells (Novagen, Darmstadt, Germany) were used as host for the heterologous expression. For cells cultivated in lysogeny broth medium at 20 °C with shaking at 180 rpm in baffled Erlenmeyer flasks, optimal protein synthesis conditions were achieved. At an optical density of OD_600_ = 0.5–0.7, overexpression of *sace_3599* was induced by addition of 0.1 mM isopropyl-β-d-thiogalactopyranoside (IPTG). The cells were cultivated at 20 °C for 20 h and harvested by centrifugation at 14,500× *g* for 20 min.

Rosetta™ 2 cells containing the plasmid pET28a-*sace_3599*-N-his_6_ were resuspended in elution buffer A (50 mM TRIS-HCl pH 8.0, 300 mM NaCl, 10 mM MgCl_2_, 10 mM imidazole) and incubated with lysozyme and DNase I on ice for 30 min. The cells were mechanically disrupted using a French press (Thermo Spectronic, Rochester, NY, USA). Subsequently, the cytosolic fraction was cleared of cellular debris by centrifugation (18,500× *g*, 20 min, 4 °C) and the protein Sace_3599 was purified by immobilized-metal affinity chromatography (IMAC) with nickel-nitrilotriacetic acid (Ni^2+^-NTA). Therefore, a 5 mL HisTrap FF column (GE Healthcare, Solingen, Germany), pre-equilibrated with elution buffer A, was loaded with the supernatant. Following extensive washing with 15% elution buffer B (50 mM TRIS-HCl pH 8.0, 300 mM NaCl, 10 mM MgCl_2_, 500 mM imidazole), 50% elution buffer B was used for the elution of the protein. Fractions containing Sace_3599 were collected and instantly concentrated to a final volume of 2.0 mL by centrifugation using a Vivaspin 20 concentrator (Sartorius Stedim Biotech, Göttingen, Germany) with a molecular mass cut-off size of 30 kDa. As a last purification step, gel filtration was performed on a size exclusion Superdex 200 10/300 GL column (GE Healthcare, Solingen, Germany), pre-equilibrated with gelfiltration buffer (25 mM TRIS-HCl pH 8.0, 300 mM NaCl, 10% [mass/vol] glycerol). Fractions containing Sace_3599 were collected, concentrated and instantly used for in vitro assays. The protein concentration was determined by measurement of the UV absorption at 280 nm.

### 4.12. In Vitro Glycosylation Activity Assay

The enzyme assays used for *N*- and *O*-Glycosyltransferase activity determination are based upon the transfer of the glucose moiety of an activated sugar donor to an acceptor catalyzed by recombinant Sace_3599 and resulting in glycosylated products. In vitro activity assays, with a final volume of 50–500 µL, were conducted at different temperatures and incubation times. Therefore, 50 µM of Sace_3599 was added to the incubation mixture containing an anthraquinone as substrate (0.1 mM), buffer (0.25 M), BSA (0.5 mg/mL), metal ions (5 mM), and a sugar donor (2 mM). A negative control without enzyme was included. After a defined incubation time and temperature, the reaction was stopped by addition of an equal volume of ethyl acetate twice. The organic phases were subsequently removed. Phases from the same reaction tubes were combined and dried in vacuo. The synthesised derivatives were analysed by HPLC-MS (see Section 4.8) and/or NMR spectroscopy (see Section 4.9).

#### 4.12.1. Variation of Time and Temperature

In vitro activity assays with U3, TRIS buffer (0.25 M; pH 8.8), MgCl_2_ (5 mM), UDP-α-d-glucose (2 mM) and a final volume of 50 µL were conducted at 20, 28 or 37 °C. After 1, 1.5, 2, 3 and 22 h reaction was stopped and the extracts were analysed as described in Section 4.12.

#### 4.12.2. Variation of Sugar Donor

In vitro activity assays with U3, TRIS buffer (0.25 M; pH 8.8), MgCl_2_ (5 mM), either UDP-α-d-glucose (2 mM) or UDP-α-d-galactose (2 mM) or TDP-α-d-glucose (2 mM) and a final volume of 50 µL were conducted at 28 °C for 22 h. The reaction was stopped and the extracts were analysed as described in Section 4.12.

#### 4.12.3. Variation of Buffer

In vitro activity assays with U3, either Citric Acid (0.25 M; pH 5.0) or PBS buffer (0.25 M; pH 6.2 or pH 6.7 or pH 7.2 or pH 7.7 or pH 8.2) or HEPES buffer (0.25 M; pH 7.0) or TRIS buffer (0.25 M; pH 7.0) or TRIS buffer (0.25 M; pH 8.8), MgCl_2_ (5 mM), UDP-α-d-glucose (2 mM) and a final volume of 50 µL were conducted at 28 °C for 22 h. The reaction was stopped and the extracts were analysed as described in Section 4.12.

#### 4.12.4. Variation of Metal Ions as Putative Cofactors

Before using the purified Sace_3599 the enzyme solution was treated with 10 mM EDTA for 1 h at room temperature. Afterwards EDTA was removed by PD-10-columns (GE-Healthcare, Solingen, Germany). In vitro activity assays with U3, TRIS buffer (0.5 M; pH 8.8), either CaCl_2_ (5 mM) or CoCl_2_ (5 mM) or MgCl_2_ (5 mM) or MnCl_2_ (5 mM), or ZnCl_2_ (5 mM) or no metal ion, UDP-α-d-glucose (2 mM) and a final volume of 50 µL were conducted at 28 °C for 22 h. The reaction was stopped and the extracts were analysed as described in Section 4.12.

#### 4.12.5. Variation of Substrate

In vitro activity assays with either U2 or U3, TRIS buffer (0.25 M; pH 8.8), MgCl_2_ (5 mM), UDP-α-d-glucose (2 mM) and a final volume of 50 µL were conducted at 28 °C. After 22 h reaction was stopped and the extracts were analysed as described in Section 4.12.

#### 4.12.6. Calculation of Conversion of Substrate to Glucosylated Product

Peaks of the mono-glucosylated products (*UG*), di-glucosylated products (*UDG*) and the substrate (*U*) of the HPLC-chromatograms of the crude extracts at detection wavelengths of 254/360 nm (wavelength/reference) were integrated. The percentage of conversion of *U* to *UG* was calculated with the help of the areas under curve (*AUC*) and the following formula:(1)$$UG\;\left[\%\right]=\frac{AUC\left[UG\right]\times 100\;\%}{AUC\left[U\right]+AUC\left[UG\right]+AUC\left[UDG\right]}$$

The percentage of conversion of *U* to *UDG* was calculated with the following formula:(2)$$UDG\;\left[\%\right]=\frac{AUC\left[UDG\right]\times 100\;\%}{AUC\left[U\right]+AUC\left[UG\right]+AUC\left[UDG\right]}$$

### 4.13. In Vitro Glycohydrolase Assay

To monitor hydrolysis of UDP-glucose by Sace_3599, different in vitro assays with a final volume of 10 µL have been performed. Sace_3599 (50 µM) was incubated at 28 °C with 10 µM UDP-[^14^C]glucose in PBS buffer (0.25 M; pH 8.2) for up to 22 h. After 1, 2 and 22 h samples of 1 µL were subjected to PEI (polyethyleneimine)-cellulose thin layer chromatography (Merck, Darmstadt, GER). LiCl (0.2 mM) was used as mobile phase to separate the hydrolysed [^14^C]glucose from the intact UDP-[^14^C]glucose. After running the samples for 12 min, the plates were dried and analysed with phosphorimager (Typhoon FLA 7000, GE Healthcare Europe, Freiburg, Germany). Quantification was performed using Multi Gauge V3.0 software.

## 5. Conclusions

We identified the new, very flexible GT Sace_3599 of *Saccharopolyspora erythraea* NRRL2338 which, in biotransformation experiments, led to α- and β-glucosylated derivatives. We showed that Sace_3599, which belongs to the inverting glycosyltransferases, is able to glucosylate aromatic amines and aromatic alcohols.

## Figures and Tables

**Figure 1 molecules-25-03400-f001:**
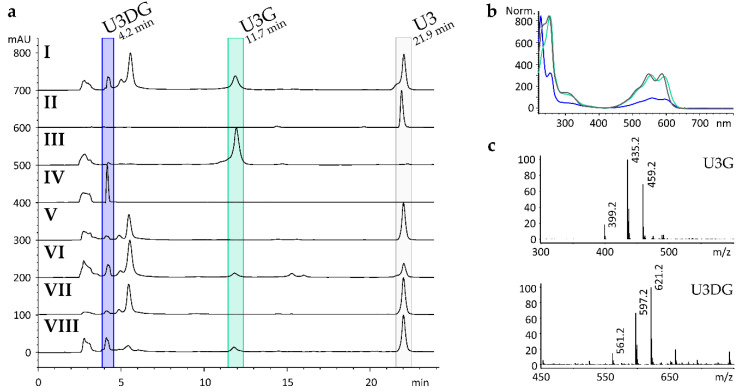
(**a**) HPLC chromatograms (λ = 254 nm) of (**I**) crude extract of *S. erythraea* fed with U3; (**II**) U3; (**III**) purified biotransformation products U3G and (**IV**) U3DG; (**V**) crude extract of *S. erythraea* Δ*sace_3599* fed with U3; (**VI**) crude extract of *S. erythraea* Δ*sace_3599* x pTOS(z)-*sace_3599* fed with U3; (**VII**) crude extract of *S. albus* Gluc fed with U3; (**VIII**) crude extract of *S. albus* Gluc x pUWL-A-*sace_3599* fed with U3. (**b**) UV-spectra of U3 (grey); U3G (turquoise) and U3DG (blue). (**c**) Mass spectra (ESI-) of U3G at 11.7 min retention time and U3DG at 4.2 min.

**Figure 2 molecules-25-03400-f002:**
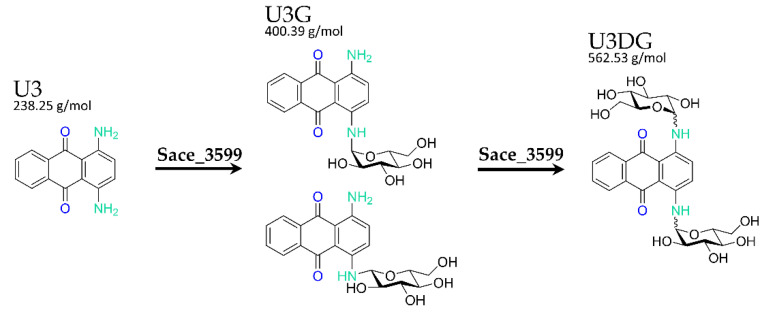
Chemical structures and masses of the substrate U3 and the corresponding mono- and di-glucosylated biotransformation products U3G and U3DG. Glycosylations are catalyzed by Sace_3599.

**Figure 3 molecules-25-03400-f003:**
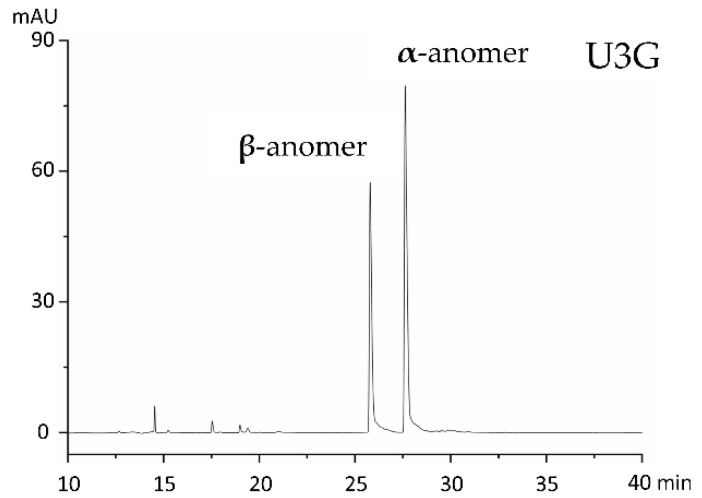
Capillary electrophoresis of U3G with separated α- and β-anomer.

**Figure 4 molecules-25-03400-f004:**
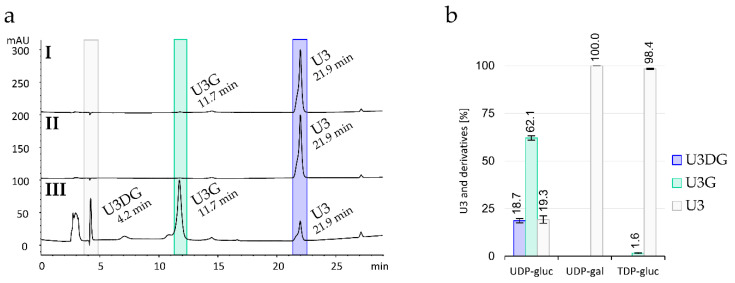
(**a**) HPLC chromatograms (λ = 254 nm) of the in vitro glycosylation activity assay with U3 and either (**I**) TDP-α-d-glucose; (**II**) UDP-α-d-galactose or (**III**) UDP-α-d-glucose used as a sugar donor. (**b**) Corresponding percentages of U3 and glycosylated products U3G and U3DG after 22 h at 28 °C. Mean values and standard deviations of three independent experiments are indicated.

**Figure 5 molecules-25-03400-f005:**
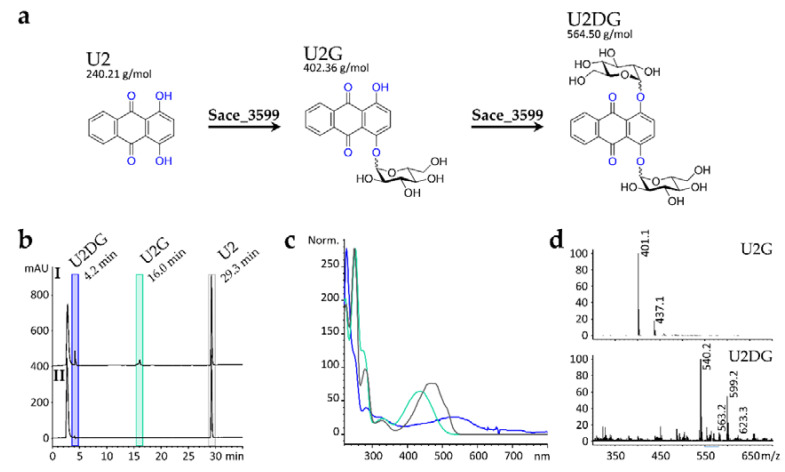
(**a**) Chemical structures and masses of the substrate U2 and the corresponding mono- and di-glucosylated biotransformation products U2G and U2DG. The glycosylations are catalyzed by Sace_3599. (**b**) HPLC chromatogram (λ = 254 nm) of the in vitro glycosylation activity assay (**I**) with U2 and UDP-α-d-glucose in comparison to negative control (**II**) without Sace_3599. (**c**) UV-spectra of U2 at retention time of 29.3 min (grey); U2G at 16.0 min (turquoise) and U2DG at 4.2 min (blue). (**d**) Mass spectra (ESI-) of U2G at 16.0 min retention time and U2DG at 4.2 min.

**Figure 6 molecules-25-03400-f006:**
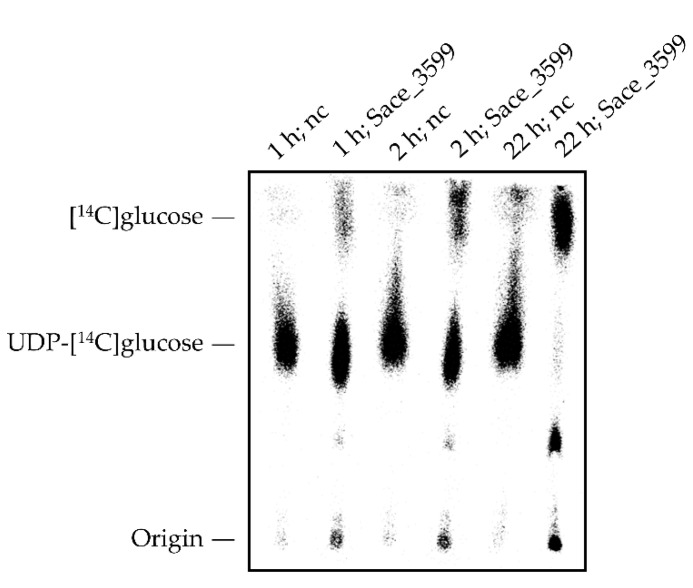
Phosphorimager picture of thin layer chromatography plate of glycohydrolase assays with Sace_3599 after 1, 2 and 22 h in contrast to negative controls (nc) without Sace_3599. A lot more free [^14^C]glucose was detected in assay-samples with Sace_3599 than in nc.

**Figure 7 molecules-25-03400-f007:**
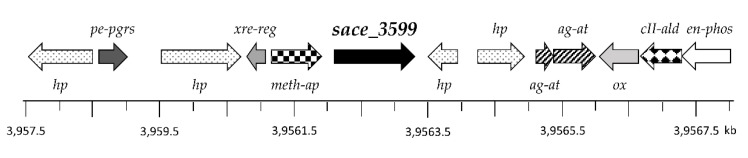
Region around the glycosyltransferase gene *sace_3599* in the genome of *S. erythraea*: *ag-at*, aminoglycoside N3′ acetyltransferase genes; *cII-ald*, gene encoding for class II aldolase; *en-phos*, gene encoding for an enolase phosphatase; *hp*, genes encoding for hypothetical proteins; *meth-ap*, methionine aminopeptidase gene; *ox*, oxidase gene; *pe-pgrs*, gene encoding a polymorphic GC-repetitive sequence family protein; *xre-reg*, gene encoding for transcriptional Xenobiotic Response Element regulator; *sace_3599*, gene encoding for glycosyltransferase.

**Figure 8 molecules-25-03400-f008:**
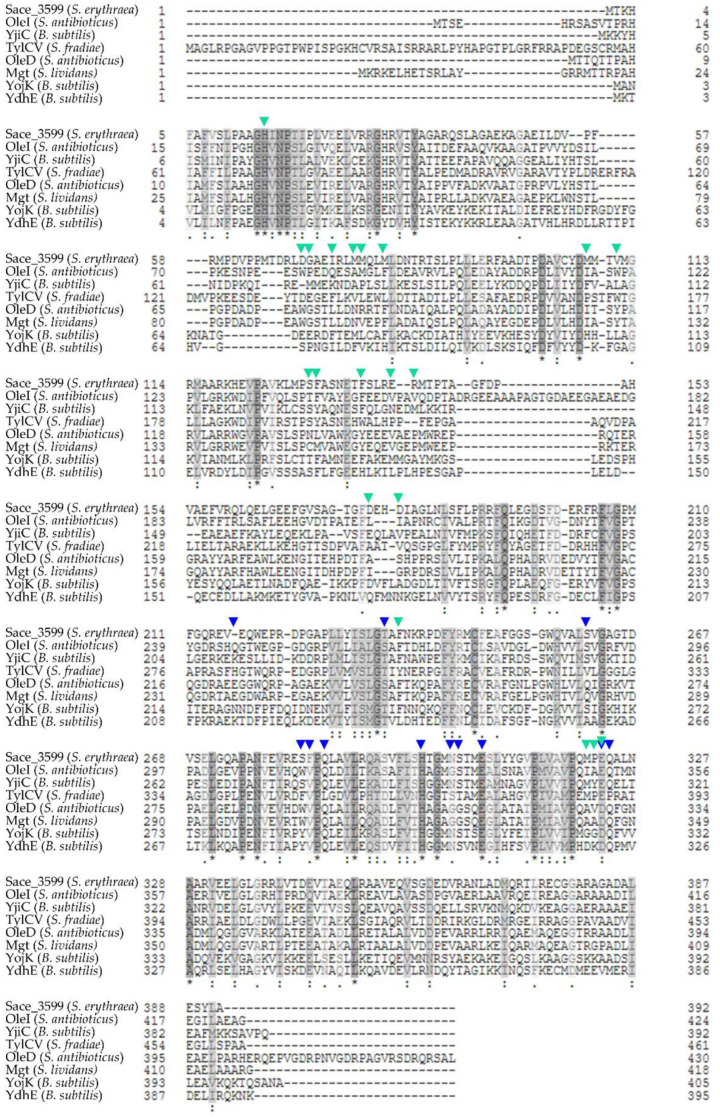
Sequence alignment of Sace_3599 and homologous proteins with similar amino acids colored in grey. Sugar donor binding amino acids (blue) and acceptor substrate binding amino acids (turquoise) of OleI are marked with arrows [42].

**Table 1 molecules-25-03400-t001:** Strains and plasmids used in this study.

Strain or Plasmid	Description ^1^	Source or Reference (s)
Strains		
*S. albus* J1074	WT strain, heterologous host	[58]
*S. albus* Gluc	WT expressing the biosynthetic gene for dTDP-d-glucose (*oleS*)	This study
*S. albus* Gluc x pUWL-A-*sace_3599*	*S. albus* Gluc expressing the gene *sace_3599*	This study
*S. erythraea* NRRL2338	WT strain, biotransformation host	[57]
*S. erythraea* Δ*sace_3599*	WT with deletion of the gene *sace_3599*	This study
*S. erythraea* Δ*sace_3599* x pTOS(z)-*sace_3599*	Deletion mutant of *sace_3599* complemented with *sace_3599*	This study
*E. coli* Turbo	General cloning host	NEB, Frankfurt am Main, Germany
*E. coli* ET12567 x pUZ8002	Strain for intergeneric conjugation	[64,65]
*E. coli* Rosetta™ 2	Heterologous expression host containing seven tRNAs for rarely used codons	Novagen, Darmstadt, Germany
Plasmids		
pTOS-Rham	pTOS derivative with *oleS*, *oleE*, *oleL*, and *oleU* under *ermE** promoter	[66]
pTOS-Gluc	pTOS derivative with *oleS* under *ermE** promoter	This study
pUWL-A [pUWL-oriT-aac(3)IV]	replicative vector for actinomycetes; *oriT*, *bla*, *aac(3)IV*, *ermE* (pUWL201)	[67]
pUWL-A-*sace_1884*	pUWL-A derivative with *sace_1884* under *ermE* promoter	This study
pUWL-A-*sace_3599*	pUWL-A derivative with *sace_3599* under *ermE* promoter	This study
pUWL-A-*sace_4470*	pUWL-A derivative with sace_4470 under ermE promoter	This study
pUWL-Dre	Replicative vector for actinomycetes; *oriT*, *bla*, *tsr*, and *dre* gene under *ermE* promoter (pUWL201)	[61]
pKC1132	Replicative vector in *E. coli*, non-replicative in actinomycetes; *lacZa*, *aac(3)IV*, and *oriT*	[68]
pLERE-spec-oriT	Cloning vector with *bla*, *aadA*, and *oriT* flanked by two *loxLE* sites and two *loxRE* sites	[66]
pKCΔ*sace_3599*	Vector for deletion of *sace_3599*, based on pKC1132	This study
pTOS(z)	integrative vector, containing *oriT*, *int*, and *attP* (VWB), aac(3)IV, *lacZ* gene, and *ermEp1** promoter	[69]
pTOS(z)-*sace_3599*	pTOS(z) derivative with *sace_3599*	This study
pBluescript II SK(−)	cloning vector for *E. coli*, with a MCS (SacI to KpnI) flanked by T3 and T7 RNA polymerase promoters; *lacZ* gene, lac promoter, pUC origin, *bla*	Agilent Technologies, Santa Clara, CA, USA
pET28a(+)	Expression vector with *aphII* and N-terminal hexahistidine affinity tag	Novagen, Darmstadt, Germany
pET28a-*sace_3599*-*N*-his_6_	pET28a(+) derivative for expression of *sace_3599*	This study

^1^*aac(3)IV*, apramycin resistance-conferring gene; *aadA*, spectinomycin resistance-conferring gene; *aphII*, kanamycin resistance-conferring gene; *attP*, attachment site on plasmid for phage integration; *bla*, beta lactam antibiotics resistance-conferring gene; *dre*, gene encoding Dre recombinase; *ermE*, constitutive promoter in streptomycetes; *ermE**, *ermEp1**, weaker variants of *ermE* promoter [70]; *int*, phage integrase gene; lac promoter, promoter of the lac operon required for the metabolism of lactose; *lacZ*, gene of the lac operon encoding for β-galactosidase; *lacZa*, variant of *lacZ* with a different MCS; *loxLE*, *loxRE*, recognition sites for Cre recombinase containing mutations within the inverted repeats; MCS, multiple cloning site; *oleE*, dTDP-glucose 4,6-dehydratase gene; *oleL*, dTDP-4-keto-6-deoxyglucose 3,5-epimerase gene; *oleS*, dTDP-d-glucose synthase; *oleU*, dTDP-4-ketohexulose reductase; *oriT*, origin of transfer; pUC origin, derivative of the pBR322 origin of replication; *rox*, recognition site for Dre recombinase; *sace_3599*, gene encoding for the *N*-GT from *S. erythraea* belonging to CAZy family 1; T3 promoter, promoter for bacteriophage T3 RNA polymerase; T7 promoter, promoter for bacteriophage T7 RNA polymerase; *tsr*, thiostreptone resistance-conferring gene.

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
