# Peer review of "Identification and Characterization of a Novel *N*- and *O*-Glycosyltransferase from *Saccharopolyspora erythraea"

_molecules, 2020, doi:10.3390/molecules25153400_

Round 1
Reviewer 1 Report
The authors investigated a novel N- and O-glycosyltransferase from Saccharopolyspora erythraea. This is an excellent report and the topic is interesting. To improve the quality of the manuscript, the authors are recommended to discuss the novelty of their works either in the section of Background or in the section of Conclusion.
- Introduction part, it is recommended to give a brief background introduction to Saccharopolyspora erythraea and why it is selected, what are its advantages and disadvantages.
- The quality of figures is low. Please provide clear figures.
- In 4.12.1. part, is the reaction interval between 3 h and 22 h too large?
- Line 162, indent.
- Line 196 should be "therefore".
- Line 396, “mL” change to “ml”. To keep the full text unified.
- Line 428-430, Can PBS buffers of different pH be integrated?
- Line 395, unit of centrifugation should be " g".
Reviewer 2 Report
It was a pleasure to read the well-prepared manuscript and I congratulate the authors for such nice results.
Two typos I could identify:
Line 196 - "Therefore, " instead of "Therefor" and Line 235 - Missing Oxford comma.
A general remark on the design: (Line 320) The authors describe that they added 25 mg/ml of each test compound for the biotransformation. As all tested compounds have a similar molecular weight, this is fine. If other compounds would be used, such an approach might be misleading. I recommend working with molarities (mM) instead of classic concentration (mg/ml).
A general remark on the chemical characterization: The authors might want to add the UV-Vis spectra (with epsilon values) and IR spectra with interpretation of the isolated compounds to increase the general impact of the paper. Furthermore, the IR spectra might support the characterization of U3DG by the absence of the NH2 typical vibrations.
Overall, I strongly recommend to publish this work in Molecules.
Reviewer 3 Report
In this manuscript Gutacker et al. identify and characterize a new N-glycosyltransferase from Saccharopolyspora erythraea. They show that the identified enzyme catalyzes the transfer of sugar moieties from a sugar donor ( dTDP-D-glucose) to an acceptor (1,4-diaminoanthraquinone, U3), producing both U3G and U3DG molecules. The manuscript is well written except for the introduction section. The introduction, indeed, does not help the reader to be well introduced into the research field of N-glycosyltransferases. The first paragraph of the introduction (line 40-43), for example, should be re-written and improved. “Sugar residues” should be changed in “sugar moieties”. Moreover, the authors should clarify what they mean with “sugars, are essential for the biological activity and pharmacological properties of secondary metabolites such as antibiotics, antimycotics, and cytotoxic drugs”. Antibiotics are not secondary metabolites.
I have some concerns about the data in some places:
- Figure 1: the authors show that U3G and U3DG products are present in crude extract after feeding the erythraea with U3. Many other sub-products are present at different retention times, such as those present at 3min and at 5.5min, Figure 1a I, V, and VII. Did the authors characterize them?
- The authors measured the glycohydrolase activity Sace_3599 protein, lines 196-198. The authors should explain how they perform this experiment and what they want to show. This data is not well discussed and needs to be framed into the context of the manuscript. Which is the added value of this data for the manuscript?
- The manuscript misses of a bioinformatics analysis. BLAST program could be used to search for protein having homology to the Sace_3599 protein. The most different sequence within those already annotated as glycosyltransferases could be discussed. Sequence alignments can be also performed. This analysis could be also used to discuss the reason why higher Sace_3599 activity has been detected when Mn2+ or Mg2+ were added. Does the homologous proteins from other organisms contain possible metal-binding sites?
Minor point
- Change “therefor” in “therefore”.
Round 2
Reviewer 3 Report
In this second version of the manuscript "Identification and characterization of a novel N- and O-glycosyltransferase from Saccharopolyspora erythraea ” the authors improved significantly the quality of the paper. They changed the first paragraph of the introduction, which is now more clear.
They also report the bioinformatics analysis of the Sace_3599 sequence. I suggest also to show a figure reporting the sequence alignment highlighting the residues belonging to the catalytic site and those involved in Mn2+ or Mg2+ binding.
Author Response
In this second version of the manuscript "Identification and characterization of a novel N- and O-glycosyltransferase from Saccharopolyspora erythraea” the authors improved significantly the quality of the paper. They changed the first paragraph of the introduction, which is now more clear.
Thank you for this nice feedback.
They also report the bioinformatics analysis of the Sace_3599 sequence. I suggest also to show a figure reporting the sequence alignment highlighting the residues belonging to the catalytic site and those involved in Mn2+ or Mg2+ binding.
Changes have been made:
Lines 230-236 & new, additional Figure 8